# Geranii Herba as a Potential Inhibitor of SARS-CoV-2 Main 3CL^pro^, Spike RBD, and Regulation of Unfolded Protein Response: An In Silico Approach

**DOI:** 10.3390/antibiotics9120863

**Published:** 2020-12-03

**Authors:** Selvaraj Arokiyaraj, Antony Stalin, Balakrishnan Senthamarai Kannan, Hakdong Shin

**Affiliations:** 1Department of Food Science & Biotechnology, Sejong University, Seoul 05006, Korea; arokiyaraj16@gmail.com; 2State Key Laboratory of Subtropical Silviculture, Department of Traditional Chinese Medicine, Zhejiang A&F University, Hangzhou 311300, China; 3Department of Chemistry, Tirunelveli Dakshina Mara Nadar Sangam (TDMNS) College, Valliyur, Tirunelveli 627113, Tamil Nadu, India; blotuskannan@gmail.com

**Keywords:** Geranii Herba, SARS-CoV-2, polyphenols, docking, GRP78, 3CL^pro^, RBD

## Abstract

Background: Since the first patient identified with SARS-CoV-2 symptoms in December 2019, the trend of a spreading coronavirus disease 2019 (COVID-19) infection has remained to date. As for now, there is an urgent need to develop novel drugs or vaccines for the SARS-CoV-2 virus. Methods: Polyphenolic compounds have potential as drug candidates for various diseases, including viral infections. In this study, polyphenolic compounds contained in Geranii Herba were chosen for an in silico approach. The SARS-CoV-2 receptor-binding domain (RBD), 3CL^pro^ (Replicase polyprotein 1ab), and the cell surface receptor glucose-regulated protein 78 (GRP78) were chosen as target proteins. Results: Based on the molecular docking analysis, ellagic acid, gallic acid, geraniin, kaempferitrin, kaempferol, and quercetin showed significant binding interactions with the target proteins. Besides, the molecular dynamic simulation studies support Geranii Herba’s inhibition efficiency on the SARS-CoV-2 RBD. We assume that the active compounds in Geranii Herba might inhibit SARS-CoV-2 cell entry through the ACE2 receptor and inhibit the proteolytic process. Besides, these compounds may help to regulate the cell signaling under the unfolded protein response in endoplasmic reticulum stress through the binding with GRP78 and avoid the SARS-CoV-2 interaction. Conclusions: Hence, the compounds present in Geranii Herba could be used as possible drug candidates for the prevention/treatment of SARS-CoV-2 infection.

## 1. Introduction

According to the World Health Organization (WHO), the coronavirus (COVID-19) infection crossed the number of 30,675,675 people worldwide and caused 954,417 deaths as of 21 September 2020. The disease SARS-CoV-2 has been classified as a β-CoV of group 2B by WHO [1]. There are seven human coronavirus (HCoV) strains that have been identified so far, categorized into α-CoV (229E and NL63) and β-CoV (OC43, HKU1, SARS, MERS, and COVID-19 HCoVs) [2,3]. Among these, MERS HCoV and SARS were reported to be more virulent and have the highest mortality [4]. HCoV is a positive-sense virus with a single-stranded 30,000 bp RNA (+ssRNA). The virus is comprised of two clusters of proteins, namely (a) the non-structural RNA-dependent RNA polymerase (RdRP), which is significant in virion replication, and the 3C-like protease (3CL^pro^) enzyme that cleaves the two polyproteins (PP1A and PP1AB) translated from viral RNA in the host cell, and (b) spike proteins to help in the fusion and entry of the virus into the host, nucleocapsid, matrix, and envelope proteins [5]. The spike (S) protein of SARS-CoV-2 binds ACE-2 through its receptor-binding domain (RBD) and the RBD-up confirmation of the S protein is a prerequisite for the formation of the RBD–ACE-2 complex [6]. The spike protein was selected for drug development instead of the ACE-2 receptor since compounds that block the ACE-2 receptor are known to have a modulatory effect on blood pressure and several other cardiovascular system-related side effects [7,8].

SARS-CoV-2 follows a similar spike protein fusion mechanism to SAR-COV. A study of a spike protein sequence analysis of SARS-CoV-2 revealed that the receptor-binding domain (RBD) is located at 319–541aa (amino acids), in which receptor-binding motifs (RBM) 437–508aa exist in the S1 subunit (14–685aa) along with the N-terminal domain (aa14–305). Especially, the major amino acids such as Thr27, Phe28, Lys31, His34, Tyr41, Lys353, Gly354, Asp355, and Arg357 that are present in the RBD help to form a network of interactions with the ACE2 receptor [9,10]. In the S2 subunit of the 686–1273aa domain, the fusion peptide is aligned from 788 to 806aa, HR1 at 912–984aa, HR2 at 1163–1213aa, the trans-membrane domain from 1214 to 1237aa, and the cytoplasm domain at 1238–1273aa [8]. The spike protein helps the virus to interact with the host cell through the inhibition of consensus protein-1 (CP-1) which is derived from the HR2 region [11].

The other possible targeting site to treat HCoV would be the 3CL^pro^ enzyme, a C30 endopeptidase cysteine protease non-structural protein [12]. The mechanism of 3CL^pro^ was deciphered computationally as the stearic interaction with glycine in polyproteins and formed a strong hydrogen bond to stabilize the complex. A conserved GSCGS (Gly-Ser-Cys-Gly-Ser) motif has been observed to form three consecutive turns, which were temporarily stabilized by a partial negative charge cluster (PNCC). This stabilizing PNCC is located on the surface opposite the active site and hence can be a potential drug targeting site for 3CL^pro^ inhibitors [13].

Considering these prototypes to target SARS-CoV-2, researchers have identified several therapeutic agents, including Western and traditional medicines, which provided an immediate clinical positive response against SARS-CoV-2 [14]. The anti-viral agents recognized and approved to date are tabulated in Table 1 [15].

However, these therapeutic drugs have severe side effects on pre-existing medical conditions in patients. Hence, natural compounds are being extensively explored by scientists across the globe [16,17,18,19]. Some of the phytochemical compounds that have been identified to possess SARS-CoV-2-inhibiting properties include 5,7,3′,4′-Tetrahydroxy-2′-(3,3-dimethylallyl) isoflavone (*Psorothamnus arborescens*), 3,5,7,3′,4′,5′-hexahydroxy flavanone-3-O-beta-d-glucopyranoside (*Phyllanthus emblica*), Myricetin 3-O-beta-D-glucopyranoside (*Camellia sinensis*), Calceolarioside B (*Fraxinus sieboldiana*), Myricitrin (*Myrica cerifera*), Methyl rosmarinate (*Hyptis atrorubens Poit*), Licoleafol (*Gycyrrhiza uralensis*), Amaranthine (*Amaranthus tricolor*), and other natural products such as Nelfinavir, Prulifloxacin, and Colistin [19].

Natural compounds are known to be bio-compatible and produce minimal or no side effects. We have strategically chosen natural compounds containing polyphenolic compounds because polyphenols have the capacity to behave as antioxidants, anticancer agents, antidiabetic drugs, anti-cardiovascular agents, and anti-neurodegenerative agents [20]. In the current study, nine natural compounds (corilagin, ellagic acid, gallic acid, geraniin, kaempferitrin, kaempferol 7-*O*-rhamnoside, kaempferol, protocatechuic acid, and quercetin) present in Geranii Herba (*Geraniaceae*) were chosen for the anti-SARS-CoV-2 in silico study. Significantly, polyphenols containing compounds are potential drugs for various viral infections. Polyphenols have a gaining momentum of activities against a virus that depends on the time point of treatment. Polyphenolic compounds are used in different stages as pretreatment, co-treatment, and post-treatment of polyphenols to inhibit the virus [21,22,23]. Based on precautionary actions, the intake of polyphenol compounds may effectively prevent viral entry. Polyphenols have virucidal effects that disintegrate the viral capsid or genome in co-treatment. In post-treatment, polyphenols exhibit their therapeutic impact in association with viral replication, assembly, and release. Plant compounds were reported as having high antiviral activity against influenza A strain and influenza B strain [24], though there is no report available on an anti-SARS-CoV-2 in silico study. Therefore, the present study was conducted to obtain structural insight into the SARS-CoV-2 RBD and 3CL^pro^ and analyze the regulation of cell signaling under the unfolded protein response in endoplasmic reticulum stress through the binding with GRP78 to avoid the SARS-CoV-2 RBD interaction.

## 2. Results and Discussion

### 2.1. SARS-CoV-2 Spike RBD

The docking analysis of polyphenolic compounds from Geranii Herba with the SARS-CoV-2-S RBD exhibited different interactions with the change in binding energies (Table 2), and the binding interactions are shown in Figure 1. Geraniin interacted with amino acid residues THR345, ARG346, SER349, LEU441, ASP442, and ASN450 particularly in the region of the SARS-CoV-2-S RBD. The ligand efficiency of geraniin is 0.31, indicating the contributions of H-bonding with different amino acids with a binding energy of −7.58 kcal/mol and van der Waals interactions with an energy of −8.86 kcal/mol. Kaempferitrin acquired an efficiency of 0.28 due to the integration of H-bonding with various amino acid residues, namely THR345, PHE347, LEU441, ASP442, ASN450, TYR451, and ARG509, of the SARS-CoV-2-S RBD, and van der Waals interactions with an energy of −9.24 kcal/mol. It has been found that quercetin can interact with VP24 to suppress its inhibition effects on IFN-I signaling. Consequently, quercetin could enhance immune activity against the Ebola virus [25]. Similarly, both quercetin and kaempferol acquired the same ligand efficiency of 0.26, whereas quercetin and kaempferol possessed slight variation in their respective binding energies of −5.71 and 5.69 kcal/mol. This difference in binding energies is due to quercetin interacting with amino acid residues SER349, LEU441, ASP442, ASN448, ASN450, and ARG509, and kaempferol interacting with residues SER349, LEU441, ASN450, TYR451, and ARG509 of the SARS-CoV-2-S. The ligand efficiency of gallic acid is 0.25, which is attributed to gallic acid interacting through H-bonding with the residues VAL341, ARG346, ASN354, and SER399 with an energy of −4.21 kcal/mol and the van der Waals interactions with an energy of −5.01 kcal/mol of the SARS-CoV-2-S. Both ellagic acid and protocatechuic acid obtained the same ligand efficiency of 0.24. Ellagic acid stimulated the immune functions, and this compound is proposed to be used along with other drugs for the chemotherapy treatment of prostate cancer patients. [26].

Ellagic acid interacted through H-bonding with the residues GLU340, ASN343, ARG346, ALA348, ASN354, and SER399 with an energy of −5.22 kcal/mol and the van der Waals interactions with an energy of −6.21 kcal/mol of the SARS-CoV-2-S. Protocatechuic acid interacted through H-bonding with the residues VAL341, ARG346, and SER399 with an energy of −4.18 kcal/mol and the van der Waals interactions with an energy of −4.53 kcal/mol of the SARS-CoV-2-S. Kaempferol 7-*O*-rhamnoside interacted through H-bonding with THR345, PHE347, SER349, and ASN450 with an energy of −5.69 kcal/mol and the van der Waals interactions with an energy of −8.27kcal/mol of the SARS-CoV-2-S. Corilagin interacted through H-bonding with the residues ARG346, PHE347, LEU441, ASP442, LYS444, and ASN450 with an energy of −3.62 kcal/mol and the van der Waals interactions with an energy of −7.43 kcal/mol of the SARS-CoV-2-S. As a result, kaempferol 7-O-rhamnoside and corilagin achieved ligand efficiencies of 0.18 and 0.08, respectively. Similar to our study, the RBD–ACE2 interaction showed strong H-bonding at TYR 505, indicating an initial contact point with ACE-2 and THR 500, ASN 501, GLY 502, TYR 449, GLN 493, and GLN 498, suggesting a strong polar interaction [27]. From this analysis, we determined that the polyphenolic compounds interact with the active sites of the SARS-CoV-2 RBD and lead to inhibiting the interaction of the SARS-CoV-2 RBD with ACE2.

### 2.2. CL^pro^, Main Protease

The gastroenteritis coronavirus M^pro^ could be involved in the self-processing that appeared at trans. It possesses active sites at cys144 and His41 and is part of a chymotrypsin-like fold that is connected by a 16-residue loop to an extra domain featuring a novel α-helical fold [28]. The sequence alignment proved that the genome sequence of SARS-CoV-2 is highly identical to SARS-CoV [18]. It was identified that there are 11 specific proteolytic sites of 3CL^pro^ at the C-terminus, forming 15 non-structural proteins. 3CL^pro^ is also called Nsp5, and it undergoes a cascade process to establish them. Initially, 3CL endures the auto-scission from the polyproteins transformed into mature enzymes, which extends the process of cleavage downstream Nsps at 11 sites to produce Nsp4-Nsp16. 3CL^pro^ directly mediates the maturation of Nsps, which is essential in the life cycle of the virus [28,29]. The profound investigations have shown the 3CL^pro^ monomer is comprised of three domains, described as domain I (residues 8–101), domain II (residues 102–184), and domain III (residues 201–303), and a long loop (residues 185–200) connects domains II and III. It has been found that the active sites Cys145 and His41 are occupied in between domains I and II, and the formation of catalytic dyad CysHis enables the nucleophile cysteine-catalyzed proteolytic process [30,31]. In the previous study, 7-hydroxyisoflavone effectively inhibited not only EV71 replication but also viral protein synthesis in a dose-dependent manner [32].

In our study, the docking analysis of a series of polyphenolic compounds shows significant interactions with 3CL^pro^ to inhibit the proteolytic process. The corilagin hydrogen bonded with amino acids THR26, PHE140, HIS164, and GLU166 of 3CL^pro^ with a binding energy of −5.82 kcal/mol. Kaempferol 7-O-rhamnoside was involved in H-bonding with LEU141, SER144, CYS145, HIS163, and PRO168 with a binding energy of −5.87 kcal/mol. Kaempferitrin committed to H-bonding with LEU141, ASN142, SER144, HIS163, and GLU166 with a binding energy of −7.83 kcal/mol, and its ligand efficiency is 0.28. Ellagic acid showed active hydrogen bonding with amino acids LEU141, GLY143, SER144, CYS145, HIS163, GLU166, HIS172, and GLN189 of 3CL^pro^ with a binding energy of −6.37 kcal/mol. Quercetin was involved in interactions with amino acids LEU141, GLY143, SER144, and GLU166 with a binding energy of −6.49 kcal/mol, and its ligand efficiency is 0.30. Gallic acid committed to H-bonding with amino acids LEU141, GLY143, SER144, HIS163, and GLU166 of 3CL^pro^ with a binding energy of −4.46 kcal/mol. Protocatechuic acid exhibited H-bonding with amino acids LEU141, GLY143, SER144, CYS145, HIS163, and GLU166 with a binding energy of −4.32 kcal/mol. Kaempferol showed a ligand efficiency of 0.36 because it was involved in H-bonding with amino acids GLY143, SER144, GLU166, and ASP187 of 3CL^pro^ with a binding energy of −7.76 kcal/mole. Geraniin was engaged in H-bonding with amino acids PHE140, LEU141, HIS163, GLU166, ARG188, GLN189, THR190, and GLN192 of 3CL^pro^ with a binding energy of −9.78 kcal/mol. Consequently, these interactions reflect in its ligand efficiency of 0.38, followed by inhibiting the proteolytic process. The compounds are arranged in an increasing tendency of binding affinity towards 3CL^pro^ such as corilagin, kaempferol 7-*O*-rhamnoside, kaempferitrin, ellagic acid, quercetin, gallic acid, protocatechuic acid, kaempferol, and geraniin, which in parallel show the increasing least binding energies.

The scrutiny of Figure 2 shows that the compounds like geraniin, kaempferol, gallic acid, quercetin, kaempferitrin, and corilagin bound the amino acid residues at the proximity of CYS145 rather than the active sites. In the docking analysis, the compounds protocatechuic acid, ellagic acid, and kaempferol 7-O-rhamnoside are directly involved in H-bonding with the active-site CYS145 along with other amino acid residues (Figure 2). This direct or indirect mode of action suggests that the compounds would envelop the active-site CYS145 to stop the further nucleophilic attack toward His41 to avoid the proteolytic process in 3CL^pro^. The obtained compound and protein interactions are shown in Table 3.

### 2.3. Glucose-Regulated Protein 78 (GRP78)

Naïve cell surfaces do not possess GRP78, whereas it exists on the surface of activated macrophages [33]. It has been found that GRP78 is the master chaperon protein to orchestrate the cell response under unfolded protein response (UPR) load stress. GRP78 resides in the lumen of the endoplasmic reticulum that communicates to keep inactivating other enzymes including activating transcription factor 6 (ATF6), protein kinase RNA-like endoplasmic reticulum kinase (PERK), and inositol requiring enzyme 1 (IRE1) [34,35]. In the infected viral cell, SAR-CoV-2 enables continuous ER stress, which generates the unfolded protein. Unprecedently, the unfolded protein response stimulates the apoptosis cell death. Under cell stress, the unfolded protein accumulation would enable GRP78 to inhibit the protein synthesis and enhancement of refolding through releasing ATF6, PERK, and IRE1. Besides, the viral infection would exert continuous stress on the ER that is responsible for translocating GRP78 to the cell surface. In addition to ACE2, the presence of GRP78 would enhance the affinity to SAR-CoV-2 of the cell. These combined factors could enable the virus to enter the host cell via a strong and stabilized mode of action. A report demonstrated that Pep42 could bind to glucose-regulated protein 78 (GRP78) selectively, internalize into highly metastatic melanoma cells, and eventually remain inside the cell [36]. Previous work has disclosed the active-site residues such as ILE426, THR428, VAL429, VAL432, THR434, PHE451, SER452, VAL457, THR458, and ILE459 [37]. These putative active sites facilitate strong interactions with the spike protein of SAR-CoV-2. Hence, our drug should interact with these active sites of GRP78 to prevent the interactions of the spike protein of SAR-CoV-2.

From docking studies, it has been identified that geraniin interacted with amino residues such as GLY430, SER452, THR456, and THR458 of GRP78 with a binding energy of −9.55 kcal/mol. Besides, geraniin acquired the inhibition constant of 100.03 nM and its ligand efficiency is 0.34. In another instance, docking revealed that ellagic acid interacted with the amino residues GLU427, VAL429, GLY430, THR458, and LYS460 of GRP78 with a binding energy of −6.47 kcal/mol. Geraniin has a more robust interaction with GRP78 than ellagic acid because both interact with different amino residues of GRP78. Kaempferol interacted with GRP78 through functional groups of kaempferol involved in hydrogen bonding with amino residues GLU427, GLY430, THR458, and LYS460, and its binding energy is −6.27 kcal/mol. The binding energies and ligand efficiencies of both ellagic acid and kaempferol are close, which indicates both compounds produced the same effects. The relative binding energies are −5.52 and −5.45 kcal/mol for quercetin and kaempferol 7-O-rhamnoside because both compounds have interactions with the corresponding amino residues of GRP78.

However, the little difference in binding energies indicates the interactions of quercetin with GLY430 and kaempferol 7-*O*-rhamnoside with LYS435 of GRP78. It implicates the significant inhibition effects with inhibition constants of 100.92 and 90.35 μM for kaempferol 7-*O*-rhamnoside and quercetin, respectively. It is evident that the interaction of quercetin with host cells has been engaged not only to inhibit the interaction with the spike protein, but also so it can arrest the apoptosis and cell cycle through a p53-dependent mechanism [38,39]. Del Carmen Juárez-Vázquez et al. found that kaempferitin gives immunostimulatory effects on immune responses mediated using splenocytes, macrophages, PBMC, and NK cells [40]. Kaempferitrin and corilagin obtained nearly identical binding energies and ligand efficiencies but were distinct in their inhibition effects. The corresponding inhibition constant values of kaempferitrin and corilagin are 639.8 and 606.67, which indicate the structurally drastic difference.

Consequently, their structural difference exhibited the modification in the interaction with amino residues of GRP78, as shown in Figure 3, and their binding details are given in Table 4. The relative values of binding energies and ligand efficiencies of gallic acid and protocatechuic acid are due to these two compounds interacting with the same amino residues, namely GLU427, THR458, and LYS460, of GRP78. It is possible to correlate the effective interactions with the inhibition constants. The inhibition constant value of gallic acid at 4.45 (mM) is larger than protocatechuic acid at 1.68 (mM), and this means gallic acid had a more significant interaction relative to protocatechuic acid with GRP78.

Similar to our result, several phytochemicals have been analyzed against SARS-CoV-2 proteins with the help of molecular docking and molecular dynamics analysis. For example, withaferin A from *Withania somnifera* with the glucose-regulated protein 78 (GRP78) receptor [41] and curcumin, brazilin, and galangin from *Curcuma* sp., *Citrus* sp., *Alpinia galanga*, and *Caesalpinia sappan* on the SARS-CoV-2 protease and the RBD of the spike protein are reported [42].

Geranii Herba (Geraniaceae) is a perennial plant found in Asia (Korea, China, and Japan) and has been traditionally used as an anti-diarrhetic drug in East Asia [43]. The plant has been recognized as safe to use for food additives and is listed in the Korean Pharmacopeia and Korean food.

### 2.4. Molecular Dynamics Simulation

The docking studies’ results indicate that the polyphenolic compounds present in Geranii Herba effectively interacted with all the target proteins along with a good binding score. Among them, the compound geraniin showed a significant binding affinity to the spike protein. The binding and inhibition effect of geraniin in the SARS-CoV-2 RBD was analyzed by the molecular dynamic simulation by Gromacs software. In the root mean square deviation (RMSD) analysis, the spatial alterations between the apo 6M17 and geraniin–6M17 complex were identified (Figure 4). When compared to the apo 6M17, the geraniin–6M17 complex has an overall low RMSD value. The deviation increased a little bit in the initial time and reduced later in the system. After 5 ns, the complex did not fluctuate more than the apo 6M17 and a sudden deviation occurred around 12 ns and continued until the end of the 20 ns. This may have happened due to the capping loop’s flexibility (472–488 residues) in the SARS-CoV-2 RBD. The capping loop in the SARS-CoV-2 RBD has some unique amino acid alignments and is more flexible when compared with other SARS CoV structures [44]. However, the overall interface of the geraniin–6M17 complex is more stable because the compound geraniin has not deviated from the receptor-binding domain. Hence, the better binding affinity of geraniin may inhibit the SARS-CoV-2 RBD, and the flexibility of the capping loop may be suppressed.

Moreira et al. [9] found that the selected amino acids in the specific region of the SARS-CoV-2 RBD, such as (ALA348-ALA352), (PHE400-ARG403), and (ASN450-ARG454), have the most considerable stability in the RBD region. The stability is due to the coupling between the two beta-strands (β4 and β5) which exist in a hairpin loop. This arrangement helps to make close communication with loops and RBD interaction with ACE2.

The compound geraniin showed interactions with the amino acids in the specific hairpin loops such as THR345, ARG346, SER349, LEU441, ASP442, and ASN450 of the SARS-CoV-2 RBD. In the root mean square fluctuation (RMSF) analysis (Figure 5), a significant drift occurs in the region of the amino acids 350–400. It might be due to the hydrogen bond interactions of geraniin with THR345, ARG346, and SER349. Similarly, the drift is identified in the capping loop (472–488 residues) region and it is due to the hydrogen bond interaction of geraniin with LEU441, ASP442, and ASN450. Hence, the strong binding affinity of geraniin may destabilize the RBD region and block the interaction with ACE2. Besides, these findings are in agreement with Toelzer et al. [45]; they identified that the linoleic acid (LA) interaction with the RBD region causes the conformational changes in the RBD, which then affect the interaction with ACE2. Further, the lower RMSF values indicate that the concerned residues are rigid and their motion is affected due to the ligand’s presence. Simultaneously, the presence of a ligand stabilizes the protein and affects its flexibility [46].

The compactness and the structural dimension of the complex 6M17–geraniin were analyzed by the radius of gyration (Rg) (Figure 6). Due to the strong binding of geraniin with the SARS-CoV-2 RBD, the Rg values show a fluctuation in the amino acids throughout the simulation. Hence, these results show the strong inhibitory effect of the compound geraniin against SARS-CoV-2.

## 3. Materials and Methods

### 3.1. Ligand and Protein Preparation

The nine natural compounds (corilagin, ellagic acid, gallic acid, geraniin, kaempferitrin, kaempferol 7-*O*-rhamnoside, kaempferol, protocatechuic acid, and quercetin) present in Geranii Herba were chosen for our study. The compound structures were retrieved from the PubChem database (https://pubchem.ncbi.nlm.nih.gov/) and checked for their bond errors in ChemDraw Ultra 12.0. Besides, energy minimization was conducted and three-dimensional structures were developed from the PRODRG2 server (http://prodrg2.dyndns.org/) [47].

The SARS-CoV-2 receptor-binding domain (RBD) (PDB ID: 6M17), 3CL^pro^ (Replicase polyprotein 1ab) (PDB ID: 6LU7), and the cell surface receptor glucose-regulated protein 78 (GRP78) (PDB ID: 5E84) were retrieved from the PDB database. For further docking study, the RBD region (223 amino acids) alone was selected from the SARS-CoV-2 spike glycoprotein. The active binding sites were analyzed by the CastP server [48].

### 3.2. Molecular Docking

The molecular docking simulation was carried out by Autodock tools embedded with MGL Tools 1.5.6v [49]. The RBD, 3CL^pro^, and GRP78 protein structures were loaded into the software and the Kollman and Gasteiger charges were computed. Water molecules were removed from the RBD, 3CL^pro^, and GRP78 structures, and hydrogen atoms were added to the polar amino acids. Then, the active amino acids were selected and the grid boxes were constructed, with 70 × 70 × 70, 90 × 90 × 90, and 60 × 60 × 60 dimensions for the RBD, 3CL^pro^, and GRP78, respectively, along with 0.375 Å spacing. For the docking analysis, a genetic algorithm (GA) was used with a population size of 150 [50]. The least energy confirmation of each ligand was chosen for further interaction analysis. PyMol (The PyMOL Molecular Graphics System, Version 2.1.0 Schrödinger, LLC) visualized the ligand–protein interaction and the hydrophobic regions were analyzed by the PoseView (Protein plus) server [51].

### 3.3. Molecular Dynamics Simulation

Based on the docking studies, the best docked confirmation of the complex 6M17–geraniin and the apo form of the SARS-CoV-2 RBD (6M17) were subjected to molecular dynamics (MD) simulation analysis and the studies were conducted with the GROMACS 2018.6 package. The ligand topology was created by the CGenFF server using the charmm36 force field [52,53,54]. The apo 6M17 and complex 6M17–geraniin were solvated by the TIP3P water model in the dodecahedron grid box. After the solvation, the system was neutralized by the addition of counter ions (NaCl). The long-range electrostatics was calculated by the particle mesh Ewald (PME) algorithm and the steepest descent algorithm minimization utilized the process. The cut-off value of 12 Å was set to calculate the electrostatics and van der Waals energy [55]. Further, the system was energy-minimized using 50,000 minimization steps. The temperature and pressure stabilized the minimized system at 300 K by the NVT (amount of substance, volume, and temperature) and NPT (amount of substance, pressure, and temperature) equilibrations [56]. In the end, a 20 ns simulation was conducted and the dynamic coordinates of the system were saved for every 2.0 fs for further analysis [57]. The Grace plotting tool (https://plasma-gate.weizmann.ac.il/Grace/) was used to analyze and visualize the data.

## 4. Conclusions

From this study, we understand that polyphenolic compounds can interact in different stages of the virion process. At the entry level, our polyphenolic compounds might interact with the active sites of the spike protein of SARS-CoV-2. In the second stage, the infected viral cell stimulates GRP78 due to aberrant ER stress, which is then translocated to the cell’s surface. The polyphenolic compounds can be engaged in interaction with GRP78 to regulate the cell signaling to release the ER stress and other processes. During the replication process of SARS-CoV-2, the polyphenolic compounds can interact either directly or indirectly with the Cys/His dyad of the main protease to inhibit the viral replication. The molecular dynamic simulation studies also proved Geranii Herba’s inhibition efficiency on the SARS-CoV-2 RBD. Overall, the polyphenolic compounds that exist in Geranii Herba (*Geraniaceae*) could be suitable drugs for SARS-CoV-2 infection and also for other viral infections.

## Figures and Tables

**Figure 1 antibiotics-09-00863-f001:**
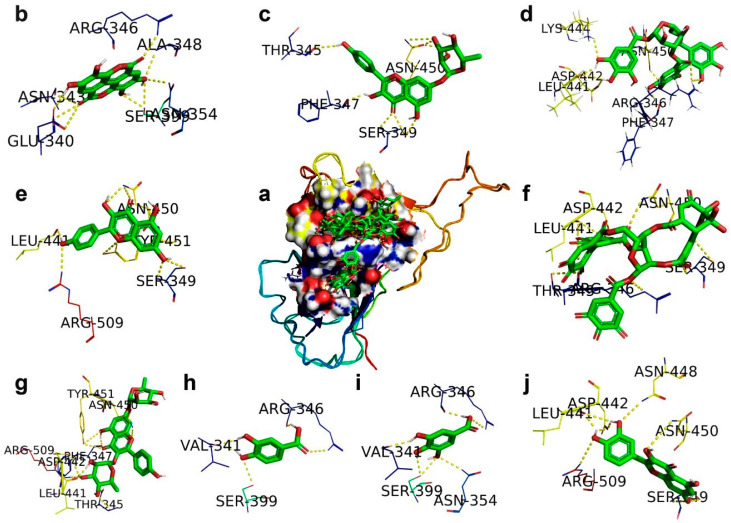
The docking mode of polyphenols present in Geranii Herba (*Geraniaceae*) with the SARS-CoV-2 spike receptor-binding domain (RBD). (**a**) The 9 polyphenols are retained in the cavity of the surface region of the RBD. (**b**) Ellagic acid. (**c**) Kaempferol 7-*O*-rhamnoside. (**d**) Corilagin. (**e**) Kaempferol. (**f**) Geraniin. (**g**) Kaempferitrin. (**h**) Protocatechuic acid. (**i**) Gallic acid. (**j**) Quercetin.

**Figure 2 antibiotics-09-00863-f002:**
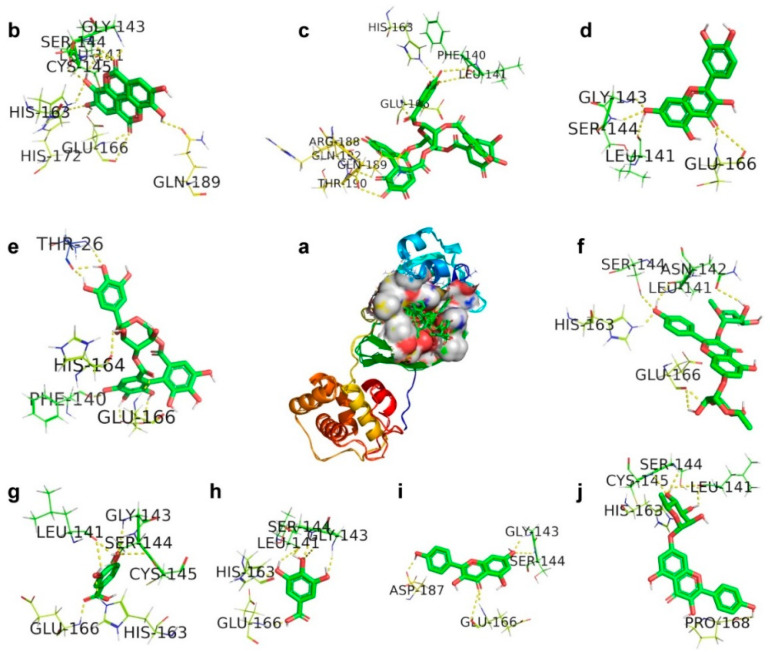
The docking mode of polyphenols present in Geranii Herba (*Geraniaceae*) with 3C-like proteinase (3CL^pro^). (**a**) The 9 polyphenols retained in the cavity of the surface region of 3CL^pro^. (**b**) Ellagic acid. (**c**) Geraniin. (**d**) Quercetin. (**e**) Corilagin. (**f**) Kaempferitrin. (**g**) Protocatechuic acid. (**h**) Gallic acid. (**i**) Kaempferol. (**j**) Kaempferol 7-*O*-rhamnoside.

**Figure 3 antibiotics-09-00863-f003:**
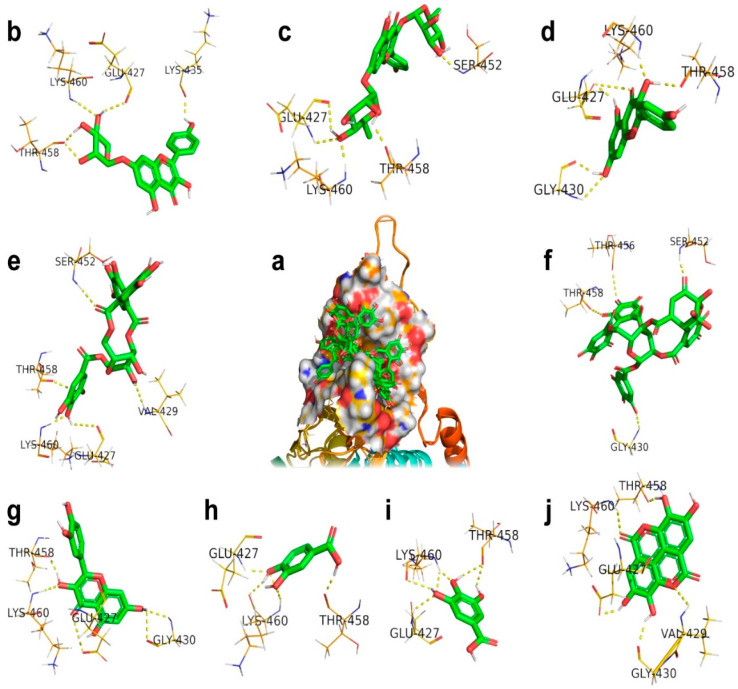
The docking mode of polyphenols present in Geranii Herba (*Geraniaceae*) with glucose-regulated protein 78 (GRP78). (**a**) The 9 polyphenols retained in the cavity of the surface region of GRP78. (**b**) Kaempferol_7_O. (**c**) Kaempferitrin. (**d**) Kaempferol. (**e**) Corilagin. (**f**) Geraniin. (**g**) Quercetin. (**h**) Protocatechuic. (**i**) Gallic acid. (**j**) Ellagic acid.

**Figure 4 antibiotics-09-00863-f004:**
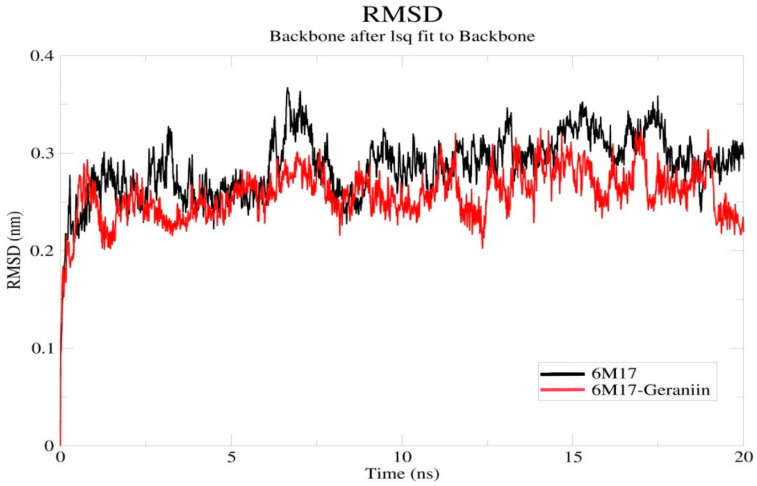
Backbone RMSDs of the apo 6M17 and 6M17–geraniin complex structures. RMSD Å (nm) is the ordinate, and time ns is the abscissa. The black and red lines indicate the 6M17 and 6M17–geraniin complex, respectively. (RMSD—Root Mean Square Deviation).

**Figure 5 antibiotics-09-00863-f005:**
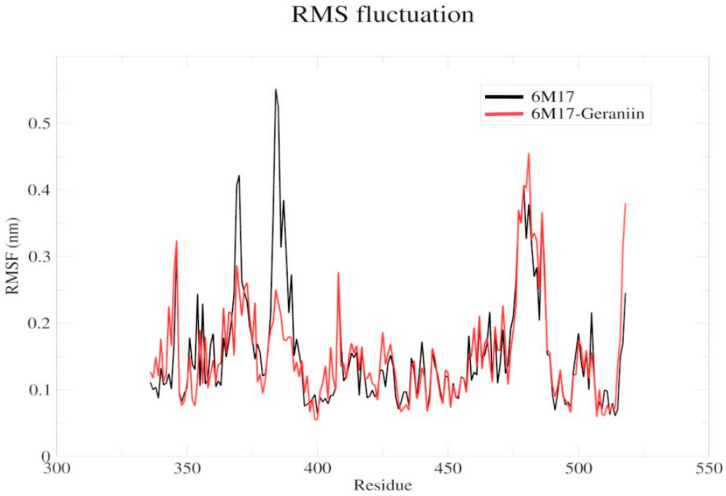
Backbone RMSFs of the apo 6M17 and 6M17–geraniin complex structures. The black and red lines indicate the comparison of the apo 6M17 and 6M17–geraniin complex, respectively. The x and y axes indicate the residue numbers and the mobility in nm, respectively. (RMSF—Root Mean Square Fluctuation).

**Figure 6 antibiotics-09-00863-f006:**
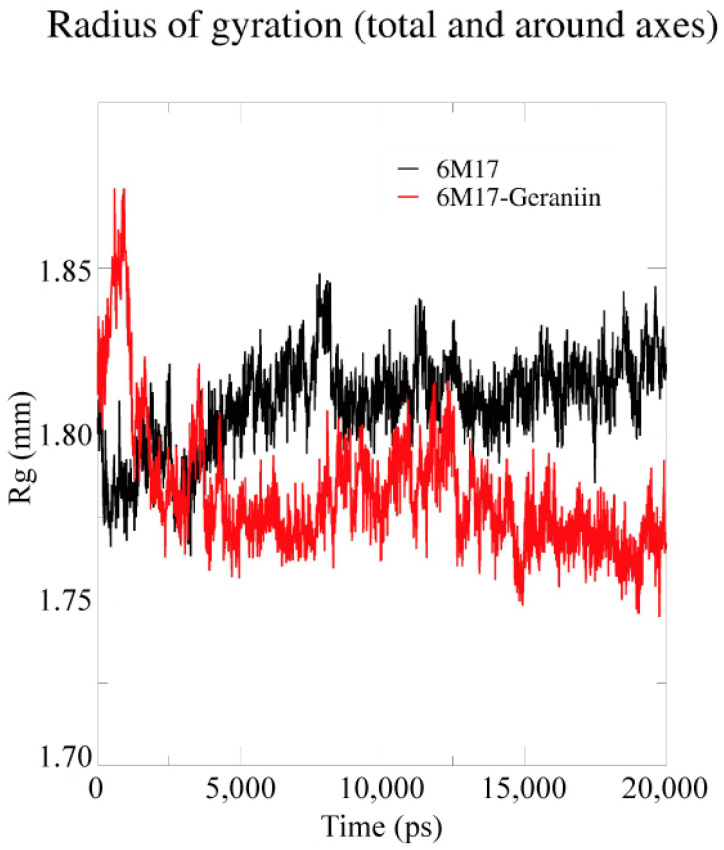
Radius of gyration (Rg) of backbone atoms of the apo 6M17 and 6M17–geraniin complex proteins. The ordinate is Rg (Å (nm)), and the abscissa is time (ns). The black and red lines indicate the apo 6M17 and 6M17–geraniin complex, respectively.

**Table 1 antibiotics-09-00863-t001:** Approved antiviral agents [15].

S.No.	Drugs	Mode of Action
1	Remdesivir	Conceals the RNA polymerase and evades viral endonuclease proofreading, thereby reducing the viral RNA production inside the host cell
2	Hydroxychloroquine and Chloroquine	Alter the pH of the lysosome, thereby deteriorating the viral proteins; also inhibit the entry of the virus into cells by interfering with the phage–host fusion through glycosylation of the ACE2 receptor and spike proteins
3	Lopinavir–Ritonavir	Restricts the protease activity
4	Umifenovir (Arbidol)	Blocks the virus–host cell membrane fusion through hydrogen bonding with phospholipids
5	Favipiravir (Avigan)	Restricts the RNA-dependent RNA polymerase
6	Oseltamivir (Tamiflu)	Inhibits the neuraminidase enzyme on the viral surface
7	Immune Interferon-alpha (IFN-α)	Inhibits the protein synthesis from viral RNA
8	Ribavirin	Interferes with the RNA metabolism

**Table 2 antibiotics-09-00863-t002:** The binding details of polyphenols present in Geranii Herba (*Geraniaceae*) with the SARS-CoV-2 spike RBD.

Ligand	Protein PDB ID	Binding Amino Acid Residues	Binding Energy (kcal/mol)	Inhibition Constant μM	VDW_HB Desolv_Energy (kcal/mol)	Ligand Efficiency
Corilagin	Spike RBD (6M17)	ARG346/1HH1, PHE347/O, LEU441/O, ASP442/OD1, LYS444/HZ1, ASN450/C	−3.62	2.23 (mM)	−7.43	0.08
Ellagic acid	Spike RBD (6M17)	GLU340/O, ASN343/O, ARG346/NH2, ALA348/N, ASN354/ND2, SER399/OG	−5.22	149.02	−6.21	0.24
Gallic acid	Spike RBD (6M17)	VAL341/O, ARG346/NH2/O, ASN354/ND2, SER399/OG	−4.21	817.47	−5.01	0.25
Geraniin	Spike RBD (6M17)	THR345/OG1/O, ARG346/NE, SER349/N.OG, LEU441/O, ASP442/OD1, ASN450/ND2	−7.58	2.79	−8.86	0.31
Kaempferitrin	Spike RBD (6M17)	THR345/OG1, PHE347/O, LEU441/O, ASP442/OD1, ASN450/ND2, TYR451/OH, ARG509/NH2,	−5.98	79.26	−9.24	0.28
Kaempferol 7-O-rhamnoside	Spike RBD (6M17)	THR345/O, PHE347/O, SER349/OG/N, 450/OD1/ND2	−5.69	66.94	−8.27	0.18
Kaempferol	Spike RBD (6M17)	SER349/OG/N, LEU441/O, ASN450/ND2, TYR451/OH, ARG509/NH2	−5.69	67.86	−7.07	0.26
Protocatechuic acid	Spike RBD (6M17)	VAL341/O, ARG346/NH2/O, SER399/OG	−4.18	857.2	−4.53	0.24
Quercetin	Spike RBD (6M17)	SER349/OD1/N, LEU441/O, ASP442/OD1, ASN448/ND2, ASN450/ND2, ARG509/NH2	−5.71	65.57	−7.16	0.26

**Table 3 antibiotics-09-00863-t003:** The binding details of polyphenols present in Geranii Herba (*Geraniaceae*) with 3C-like proteinase (3CL^pro^).

Ligand	Protein PDB ID	Binding Amino Acid Residues	Binding Energy (kcal/mol)	Inhibition Constant μM	VDW_HB Desolv_Energy (kcal/mol)	Ligand Efficiency
Corilagin	3CL^pro^ (6LU7)	THR26/HN/O, PHE140/O, HIS164/O, GLU166/HN/OE2,	−5.82	54.33	−9.44	0.13
Ellagic acid	3CL^pro^ (6LU7)	LEU141/O, GLY143/HN, SER144/HN/HG, CYS145/HN, HIS163/HE2, GLU166/OE2/HN/O, HIS172/HE2, GLN189/OE1,	−6.37	21.54	−7.18	0.29
Gallic acid	3CL^pro^ (6LU7)	LEU141/OGLY143/HN, SER144/HN, HIS163/HE2, GLU166/HN	−4.46	535.39	−4.86	0.32
Geraniin	3CL^pro^ (6LU7)	PHE140/O, LEU141/O, HIS163/HE2, GLU166/HN, ARG188/O, GLN189/O, THR190/O, GLN192/1HE2	−9.78	67.61	−10.76	0.38
Kaempferitrin	3CL^pro^ (6LU7)	LEU141/O, ASN142/OD1, SER144/OG, HIS163/HE2, GLU166/O	−7.83	286.47	−8.48	0.28
Kaempferol7-O-rhamnoside	3CL^pro^ (6LU7)	LEU141/O, SER144/HN, CYS145/HN, HIS163/HE2, PRO168/O	−5.87	49.91	−8.42	0.19
Kaempferol	3CL^pro^ (6LU7)	GLY143/HN, SER144/N, GLU166/O/HN, ASP187/O	−7.76	2.04	−9.14	0.36
Protocatechuic acid	3CL^pro^ (6LU7)	LEU141/O, GLY143/HN, SER144/HN, CYS145/HN, HIS163/HE2, GLU166/HN	−4.32	677.36	−4.59	0.34
Quercetin	3CL^pro^ (6LU7)	LEU141/O, GLY143/HN, SER144/HN, GLU166/HN	−6.49	3.24	−9.2	0.30

**Table 4 antibiotics-09-00863-t004:** The binding details of polyphenols present in Geranii Herba (*Geraniaceae*) with glucose-regulated protein 78 (GRP78).

Ligand	Protein PDB ID	Binding Amino Acid Residues	Binding Energy (kcal/mol)	Inhibition Constant μM	VDW_HB Desolv_Energy (kcal/mol)	Ligand Efficiency
Corilagin	GRP78(5E84)	GLU427/O, VAL429/HN, SER452/HN, THR458/O, LYS460/HN	−4.39	606.67	−8.32	0.10
Ellagic acid	GRP78(5E84)	GLU427/OE2, VAL429/HN, GLY430/O, THR458/O, LYS460/HN	−6.47	18.05	−7.24	0.29
Gallic acid	GRP78(5E84)	GLU427/HN, THR458/O, LYS460/HN/O	−3.19	4.45 (mM)	−3.42	0.27
Geraniin	GRP78(5E84)	GLY430/HN, SER452/HN, THR456/O, THR458/O	−9.55	100.03 (nM)	−10.38	0.34
Kaempferitrin	GRP78(5E84)	GLU427/O, SER452/HN, THR458/HN/O, LYS460/HN	−4.36	639.8	−8.0	0.11
Kaempferol 7-O-rhamnoside	GRP78(5E84)	GLU427/O, LYS435/O, THR458/O, LYS460/HN	−5.45	100.92	−7.91	0.18
Kaempferol	GRP78(5E84)	GLU427/OE1, GLY430/HN, THR458/O, LYS460/HN	−6.27	25.23	−7.4	0.30
Protocatechuic acid	GRP78(5E84)	GLU427/HN, THR458/O, LYS460/O	−3.78	1.68 (mM)	−3.33	0.24
Quercetin	GRP78(5E84)	GLU427/OE1, GLY430/HN/O, THR458/O, LYS460/HN	−5.52	90.35	−6.98	0.25

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
