# Peer review of "Geranii Herba as a Potential Inhibitor of SARS-CoV-2 Main 3CLpro, Spike RBD, and Regulation of Unfolded Protein Response: An In Silico Approach"

_antibiotics, 2020, doi:10.3390/antibiotics9120863_

Round 1
Reviewer 1 Report
he work presented by Arokiyaraj et al is focused on the docking study of several plant derivatives from Geranii Herba with the aim to inhibit cell entry and proteolytic process. Thus, the authors have chosen a set of protein such as a spike domain that recognizes the ACE2 receptor and play a role in fusion and entry of the virus (RBD), a protein that is key for the cleavage of polyproteins (3CL pro) and the surface receptor GRP78 protein. I believe these 3 protein cases represent the complexity in viral pathogenesis from the SARS-CoV-2 recognition, cell entry , fusion, replication inside the host and externalization. In this regard, I suggest the authors make an effort to include a brief discussion of the current pathogenesis in a cartoon-like fashion and link to it the role of the chosen protein. It will facilitate readers to get an overview of the whole study.The manuscript as a whole lacks in some parts a clear description. For instance:- line 54: ..319–541 aa, and its motif at 437–508 aa in S1 subunit (14–685 aa). >>> The last part is known as the receptor binding module.
- line 57: This spike protein that helps the interactions of the virus with the host cell is inhibited through the protein CP-1 derived from the HR2 region [9]. >>> it is not clear, the CP-1 acronym has not been introduced.
- line 73: SARS-CoV >>> SARS-CoV-2 and in some part of the text the authors employ SARS-COV-2. Please unify the acronym.
- line 129: ` The ellagic acid ... >>> several " ` " after residues name, beginning of sentences, etc.
I see the authors have not made an effort to put in perspective their docking results. It is true that main goal is to inhibit, but without molecular dynamics simulation those results represent an insuffient study. Since, all-atom MD is quite demanding, I suggest to compare their results with docking regions that affect to some extend the stability of the spike. Recently, It has been published an assessment of the mechanical stability of the SARS-CoV-2 RBD (R. Moreira et al. Nanoscale, 2020, DOI: 10.1039/D0NR03969A) and the region with amino acids residues if disrupted by docked molecules can have a great effect on destabilising the RBD-ACE2 complex. Another study that deserve to be compare is binding of linoleic acid (LA) in the hydrophobic pocket of the RBD. This molecule lock the close conformation of the spike protein and disable partially transition to a pre-fusion state where the RBD transit to the up conformation and become accessible to the ACE (Toelzer et al. Science, 2020 DOI:10.1126/science.abd3255).
Furthermore, the quality of all images is quite poor. All figures need to improved before publishing.
Author Response
Dear Editor,
We thank the reviewer for their comments in improving our manuscript quality to publish in “Antibiotics”. Based on the reviewer comments we responded point by point. Changes made in the manuscript highlighted in blue color font.

Reviewer 2 Report
This article addresses an important and current topic.
The topics related to Severe Acute Respiratory Syndrome Coronavirus 2 (SARSCoV-2) have grabbed the attention of the scientific community because they may deliver new insights to treat the COVID-19 pandemic. So, it is necesary to provide and share as much information as possible to set guidelines on how to act and eliminate the virus.
The paper is worthy but needs some minor improvements before its presentation
lines 62 to 64: The terms partial negative charge cluster (PNCC) and GSCGS motif are used without previous definition even when they are terms frequently referenced.
Table 1 is located in the text without any reference. Only at line 69 this table is mentioned.
On line 52 and 53 the authors have uncertainties if SARS-COV-2 follows the mechanism Spike protein fusion, based on reference [8] and [9]. Could the authors consider if this statement agrees with other studies such as the published by Armijos-Jaramillo et al. https://doi.org/10.1111/eva.12980 and also with the discussion provided by Wanh et al. at ACS Catal. 2020, 10, 5871−5890 https://doi.org/10.1021/acscatal.0c00110
It would be convenient for the authors to extend the comparison to those published by other scientists on this matter.
Author Response
Dear Editor,
We thank the reviewer for their comments in improving our manuscript quality to publish in “Antibiotics”. Based on the reviewer comments we responded point by point. Changes made in the manuscript highlighted in blue color font.
Kind regards
Hakdong Shin
Department of Food Science and Biotechnology
Sejong University

Round 2
Reviewer 1 Report
- I appreciate the effort done by the authors to bring MD simulation as an analysis tool. It increases the chances to accept those results even-though the MD time scale is limited to 20 ns. However, the trajectory analysis looks quite stable and I will accept it as a proof of the good binding. However, I will ask the authors to improve the resolution of the MD analysis in Fig. 4-6 and make them more standard/clear.
- Use the same terminology SARS-CoV-2 and not SARS-COV
- the meaning of "aa" is not defined
- Improve quality of figure 2 (central RBD) is still low resolution, one can not see the binding places.
Author Response
Title: Geranii Herba as a potential inhibitor of SARS-CoV-2 main 3CL pro, Spike RBD and regulation of unfolded protein response: An in silico approach
Journal: Antibiotics-956390 (Minor Revision)
Responses to Reviewer 1:
I appreciate the effort done by the authors to bring MD simulation as an analysis tool. It increases the chances to accept those results even-though the MD time scale is limited to 20 ns. However, the trajectory analysis looks quite stable and I will accept it as a proof of the good binding. However, I will ask the authors to improve the resolution of the MD analysis in Fig. 4-6 and make them more standard/clear.
Ans: Based on the reviewer’s comment, the image quality of figure 4 – 6 was improved.
- Use the same terminology SARS-CoV-2 and not SARS-COV
Ans: Correction made as per the reviewer comments.
- the meaning of "aa" is not defined
Ans: Correction made as per the reviewer comments. Line no. 59 (amino acids)
- Improve quality of figure 2 (central RBD) is still low resolution, one can not see the binding places.
Ans: Based on the reviewer’s comment, the image quality of figure 2 was improved (Line no 215).
Kind regards
Prof. Hakdong Shin
Associate Professor
Department of Food Science and Biotechnology
Sejong University
South Korea